# The Impact of Heat Treatment on the Behavior of a Hot-Dip Zinc Coating Applied to Steel During Dry Friction

**DOI:** 10.3390/ma14030660

**Published:** 2021-01-31

**Authors:** Dariusz Jędrzejczyk, Elżbieta Szatkowska

**Affiliations:** 1Department of Mechanical Engineering Fundamentals, University of Bielsko-Biala, 43-309 Bielsko-Biała, Poland; 2BOSMAL Automotive Research and Development Institute Ltd, 43-300 Bielsko-Biala, Poland; ela.szatkowska@gmail.com

**Keywords:** heat treatment, friction coefficient, hot-dip zinc galvanizing, coating hardness

## Abstract

The analyzed topic refers to the wear resistance and friction coefficient changes resulting from heat treatment (HT) of a hot-dip zinc coating deposited on steel. The aim of research was to evaluate the coating behavior during dry friction after HT as a result of microstructure changes and increase the coating hardness. The HT parameters should be determined by taking into consideration, on the one hand, coating wear resistance and, on the other hand, its anticorrosion properties. A hot-dip zinc coating was deposited in industrial conditions (according EN ISO 10684) on disc-shaped samples and the chosen bolts. The achieved results were assessed on the basis of tribological tests (T11 pin-on-disc tester, Schatz^®^Analyse device, Sindelfingen, Germany), microscopic observations (with the use of optical and scanning microscopy), EDS (point and linear) analysis, and microhardness measurements. It is proved that properly applied HT of a hot-dip zinc coating results in changes in the coating’s microstructure, hardness, friction coefficient, and wear resistance.

## 1. Introduction

According to a report by the International Lead and Zinc Study Group [1], the global zinc production in 2016–2019 amounted to more than 13 million tons yearly, and more than half of these resources were used to protect steel from corrosion in galvanic processes. This is an up-to-date, highly popular, and trusted technology [2,3]. Hot-dip zinc galvanized products made of Fe–C alloys are characterized by a number of advantages, such as high corrosion resistance, relatively low production cost, and simplicity of the process. The development of the industry and, as a consequence, the growing needs of new applications determine continuous technological progress. 

The anticorrosion properties of zinc coatings are determined by the structure of the created layer, which is composed of a few sublayers—Figure 1.

Generally, there are three phases occurring in the Fe–Zn diagram [5,6,7] as a result of the peritectic reaction—Γ-Fe_3_Zn_10_, δ-FeZn_7_, and ζ-FeZn_13_—and an iron solid solution in zinc-η, which settles on the surface as it is pulled out of the bath (Figure 1a). There is controversy concerning the order of zinc coating growth. The mechanism suggested by Onishi [8] assumes the formation of phases at the liquid phase boundary, from the highest content of zinc to the highest content of iron: the ζ, δ, Γ1-model based on reactionary diffusion (the growth of new phases with crystal lattices different from the lattice of diffusing components). The proposed sequence of growth was observed for relatively long experiment times—up to 64 h. In another work [9], the author doubted that the ζ phase forms as the first one, suggesting rather that in real conditions of galvanizing, which takes only a few minutes, a very thin layer of over-cooled liquid solution forms at the iron surface, which supports gradual creation of the Γ1 phase. The coating itself grows as a result of peritectic reactions; thanks to this, δ and ζ phases are created [10].

According to the presented model [4], we can assume that in the Fe–Zn system, the Γ1 phase will be observed as the first one; next, within a few seconds, a sublayer of δ_c_ (compact) and δ_p_ (palisade) is created. The presented mechanism proves that the properties of the steel base can also influence the kinetics of Zn coating growth. In fact, the microstructure of the hot-dip zinc coating can differ significantly from that in the model. Such factors as the method of preparation of the galvanized surface [11,12,13,14,15], the chemical composition of the base surface [16,17,18], the chemical composition of the zinc bath [19,20,21,22,23,24], the parameters of galvanization [25,26,27,28,29], etc., may cause microstructural differentiation even within the same coating (Figure 1b). Most of the mentioned parameters influence the reactivity of steel, and only the optimal selection of galvanizing conditions enables the formation of a proper zinc coating. For example, when galvanizing high-silicon steels [16,30], the coatings are much thicker than allowed by the corresponding standard [12]. The reactivity of steel in the Sandelin range (steel with silicon) can be controlled by adding Ni. Control of the steel’s reactivity can be also achieved with the assistance of Al, Bi, and Sn [19]. The mechanical properties of the zinc coating are dependent on its microstructure and are consequences of the properties of the individual phases visible in the coating cross section. The hardness values of specific layers of the hot-dip zinc coating are as follows: Fe_3_Zn_10_, not determined; FeZn_7_, 270 HB (Brinell hardness); FeZn_13_, 220 HB; 100% Zn, 50–70 HB [31]. The hardness values of the most frequently used zinc coatings (hot-dip, galvanic, lamellar, sherardized) differ essentially, and the lowest values are obtained by hot-dip galvanizing. The growing requirements for various structural elements also apply to their tribological properties, which are in direct correlation with the hardness and microstructure of the applied coating [32]. There is a visible tendency to apply harder and harder coatings. Although zinc-plated coatings deposited via galvanic technology demonstrate hardness of around 50 HV (Vickers hardness), it is possible to increase this hardness level up to 500 HV by introducing alloying elements, i.e., Ni (14%) [33,34]. In the case of lamellar zinc coatings, the hardness of the outer surface can reach even 500 HV, which is due to the application of thermosetting surface varnish as the last layer. Moreover, other coatings can be used as alternative solutions: heat-treated Ni–P coating (1050 HV50) [35], or even heat-treated hard Cr coating (1700 HV 0.1) [36]. As shown in the literature review, heat treatment (HT) is a method that is more and more often applied as a way of improving different coating properties.

The data regarding HT’s influence on hot-dip zinc coatings’ properties provided in the literature are incomplete and refer to a wide range of temperatures. In the publication [37], an increase of the zinc coating hardness of more than double was reported, but no information was given about the HT parameters. The coating microstructure also changed during the formability of hot-dip galvanized DC01 steel [38]. During the experiment, temperatures in the range of 500–540 °C were applied for 10–180 s. It has been found that the use of higher temperatures in combination with shorter processing times improves the coating formability behavior. The "hot forming" process is also accompanied by a similar effect (910 °C, 300 s) [39].

The aim of this paper was to confirm that HT of a hot-dip zinc coating is a simple and economical post-bath treatment to increase the hardness and wear resistance of the coating. In addition to its influence on the microstructure, mechanical properties, and corrosion resistance, heat treatment of the zinc coating can also affect the friction coefficient—a very important design parameter. Considering the great potential of HT as a tool to fit the friction coefficient to the specific application requirements, in this experiment, we aim to broaden practical knowledge in this area.

## 2. Materials and Methods

The presented research (marked in red—Figure 2) is a part of a wider study that also concerns cast iron and more parameters influencing the analyzed results. During the experiment, different samples were used: disc-shaped samples measuring 25 mm in diameter and 4 mm in thickness made of low-carbon DC01 steel, and 23MnB4 steel bolts (M12-60, M16-100). 

Samples were hot-dip galvanized according to EN ISO 10684 [40]—a process of etching in 12% HCl, fluxing, and dipping in a Zn bath with Al (0.002%), Bi (0.055%), and Ni (0.058%), at a temperature of 460 °C within 1.5 min, followed by cooling in water. Next, samples were subjected to controlled heat treatment in the temperature range 200–530 °C. This process was carried out in an electric chamber furnace. The time of treatment was 7 min for disc-shaped samples and 11 min for bolts. The HT parameters were selected on the basis of our own preliminary studies, a literature review [37,38,39], and measurements of the heating rates of the treated elements. After the heat treatment, samples were taken out of the furnace chamber and were air-cooled to ambient temperature. In all the tests, the samples were cut manually with a saw blade and then hot-embedded in resin, grinded, and polished. The following parameters were analyzed during investigations: the heating process (chamber furnace FCF 75HM, Flir E95 thermal imaging camera, FLIR Systems, Wilsonville, OR, USA); the friction coefficient (T11 pin-on-disc tester; the vertical Schatz Analyse M12 system, type 5413-2777-03 C, Kistler Instrumente GmbH, Sindelfingen, Germany); the microstructure of the zinc coating structure and steel using an Axiovert 100 A optical microscope (Zeiss Group, Oberkochen, Germany) and an EVO 25 MA Zeiss scanning electron microscope with an EDS attachment (Zeiss Group, Oberkochen, Germany); and microhardness changes in the cross section of both the coating and the subsurface layer of steel (Vicker’s HV 0.02, Mitutoyo Micro-Vickers HM-210A device 810-401 D, Mitutoyo Corporation, Kanagawa, Japan). Advanced research techniques that are used today for the comprehensive testing of the mechanical properties of different coatings [41,42] allow simultaneous determination of the coating hardness and the modulus of elasticity at the nanoscale. Taking into account the fact that the thickness of the tested coating (80–100 µm) is much greater than that in the compared study (3 µm [41]), microhardness measurements are the most appropriate method to analyze the range of occurrence of the individual sublayers in the investigated coating. 

## 3. Results and Discussion

### 3.1. Metallographic Observations and Microhardness Distribution

Before the tribological and metallographic examinations, the zinc coating thickness was measured with the use of an electronic PosiTector 6000 tester (DeFelsko Corporation, Ogdensburg, NY, USA) (magnetic induction method, head diameter Ø = 9 mm). The thickness of the coating on the bolts’ heads, as well as the disc-shaped samples, after galvanizing and after HT each time was in range of 80–100 µm.

The changes observed in the zinc coating morphology occurred in accordance with the Fe–Zn equilibrium diagram [4,5,6]. The coating without heat treatment showed a structure composed of four phases, whereas the structure after heat treatment revealed practically three phases (there was no pure η phase). The typical microstructure of a zinc coating obtained without HT is shown in Figure 3a. The diffusion zone was formed by layers of the intermetallic phases of the Fe–Zn system: η, ζ, δ, and Γ1. The Γ1 phase is very thin (about 1 µm [43]) and is therefore difficult to identify. On the other hand, the δ phase layer is thicker and compact. The next phase, ζ, shows two distinct zones [19]. The inner layer, formed by reactive diffusion between Fe and Zn, has a stable thickness and compact structure. The outer layer is more diverse—the δ phase particles are practically embedded in the matrix of the η phase due to dissolution in liquid zinc and secondary separation. As a result, zones of loosely packed elongated crystals are formed. The outer coating layer, η, is formed when the material is removed from the bath. An increase in HT temperature results in an increase in the thickness of the iron-rich layers (Γ, δ, ζ) (Figure 3b). The structure formed after the conducted experiment is similar to the result observed in hot-formed steel samples galvanized in a zinc bath with temperature stabilized at 460 °C; samples were cooled to room temperature, reheated to 500 °C, and then cooled again at a rate of 10°/s to room temperature [39]. Although the structure of the zinc coating after the HT described in [37] was composed of four layers, most of the data [4,38,39] agree that there are only three phases in a zinc coating after heat treatment. During HT, the δ and Γ phases grow at the expense of the ζ phase [39], and at higher temperatures, the δ phase grows towards the surface of the coating, consuming the ζ layer [5]. On the basis of the achieved result, it is clear that after HT at 460 °C, there was practically no pure η phase. Under normal conditions of hot-dip zinc galvanizing, the thickness of the individual phases of the zinc coating depends on the dissolution process (going into the liquid phase) of the growing phase at the interface with liquid zinc. During HT, the conditions for the growth of individual phases are different than in the case of bath zinc galvanizing because the Zn amount in the coating is limited and the coating thickness does not change significantly. It is quite possible that the model presented in Figure 1a [4] is also valid in this case, but the occurrence of bulk diffusion resulting in thickening of the δ and ζ phases is more probable.

Figure 4 presents the averaged hardness results (average of five measurements) measured on the cross sections of galvanised disc-shaped steel samples. For the reference samples (untreated, UT), results close to the values presented in the literature were achieved [30]. Next, analysis of the heat-treated coatings showed that with the application of a higher treatment temperature, the hardness of the coatings increased. Moreover, throughout the cross section of the coating, an upward tendency was observed. Measurements made on the samples subjected to HT at 430 °C showed an increase in microhardness in the subsurface layer to approximately 2.5 times that of the reference samples. On the other hand, a 530 °C HT temperature gave a microhardness increase of approximately 5 times. The presented hardness changes are due to the diffusion of iron from the base material deep into the coating (and, consequently, changes in the microstructure), which was confirmed by additional studies (including EDS analysis). It follows from the reported results that an applied temperature close to 530 °C combined with 7 min holding time is definitely too high considering the experimental target hardness level in the range of 100–150 HV. Based on the above results, HT of zinc coatings deposited on bolts was conducted in the temperature range of 300–430 °C and compared to the result achieved at 530 °C. 

The structure observed in the coating deposited on tested bolts corresponds well with the measured microhardness distribution in the coating and subsurface steel layer cross section (Figure 5)**.** It is clear that as a result of heat treatment, the outside coating hardness essentially increased from 58 (UT) to 250 (300 °C) and 285 HV 0.02 (430 °C) (Figure 6). The hardness values measured for the coatings on the bolts were much higher than those for the coatings on the disc-shaped samples. Unfortunately, the hardness of the subsurface steel layer slightly decreased as an effect of tempering. At a distance of 20 µm from the steel surface, the measured average hardness values were as follows: 320 (for UT samples), 300 (after HT at 300 °C), and 290 HV 0.02 (after HT at 430 °C). The tempering effect was also visible at a greater depth. Measurement of the HBW hardness (Brinell hardness—a carbide ball indenter) at a depth of 2 mm confirmed that the average hardness values decreased correspondingly from about 270 (UT) to 255 (300 °C) and 245 HBW (430 °C). Microscopic examinations (via optical microscope) confirmed that there were no delaminations, cracks, or surface degradation visible as a result of the conducted heat treatment of the zinc coating.

### 3.2. Friction Coefficient Measurements 

The determination of friction characteristics was performed with the use of two methods: a T11 pin-on-disc tester and a Schatz Analyse M12 testing machine system, type 5413-2777-03 C. The pin-on-disc test was applied to study the friction and wear of heat-treated zinc coatings in sliding conditions, to determine the average coefficient of friction between a friction pair, and to evaluate the rate of wear of the disc-shaped sample surfaces [44,45,46]. The Schatz Analyse system was used only to test the selected zinc-coated bolts after HT under practical conditions and to evaluate the friction coefficient with reference to assembly processes.

#### 3.2.1. The Pin-on-Disc Method

The tribological investigations with application of the T11 device consisted of testing the steel pin/zinc coating couple in dry friction conditions and calculating the friction coefficient. To conduct the experiment, surfaces of the tested disc-shaped samples were subjected to friction with a Ø 4 steel rod, with a constant load of F = 4.9 N, which moved in circles on the surface of the samples at a rate of n = 45 rotations/min for a duration of 30 min. During the test, the friction coefficient was measured every 0.2 s.

The coatings showed higher abrasion resistance with increasing heat treatment temperature, which was manifested by a decrease in weight loss (Figure 7). This was particularly evident in the case of coatings treated at 270–430 °C. The difference in weight loss between the coatings heat-treated at 430 and 460 °C was very small (0.0007 g), and for the coating treated at 530 °C, an increase in weight loss was observed.

Macroscopic analysis of the external appearance of the coatings confirmed that after HT (in the range of 270–430 °C), the surface of the disc-shaped samples’ coatings (not exposed to the friction test) did not essentially change (in either topography or coloring). The groove resulting from the friction during the test was regular. The particles rubbed away from the coating had a "flake" shape and were 0.1–0.6 mm (Figure 8a). This confirmed the relatively high plasticity of the coatings treated in this temperature range. Starting from the 460 °C temperature level, the coating color changed from grey-silver to grey and dark grey. Additionally, the sheared zinc coating particle shape changed from “flakes” to “fine powder” (Figure 8b).

The formation of more-plastic, "flake"-shaped friction products is more favorable—it promotes a reduction in abrasive wear, while hard "fine powder" grains can act as microblades that increase coating wear. Considering the dependence of the abrasion resistance of the coatings of disc samples on the HT temperature, and the tendency to change the character of the friction product at 460 °C, 430 °C was adopted as the optimum temperature for further studies. Preliminary studies with the use of EDS showed that the change in color of the outer surface of the disc specimens’ coatings could be caused by increased iron content and, consequently, a change in the morphology of the microstructure. SEM studies (Figure 3b, Table 1) confirmed that a crystalline-plate structure was present on the coating surface and that only three phases were present in the zinc coating after HT at 460 °C: Γ (23.5–28.0 wt% Fe), δ (7.0–11.5 wt% Fe), and ζ (6.0–6.2 wt% Fe) [47,48]. A comparison of the obtained results with the literature data suggests that the ζ phase (6.18 wt% Fe) appeared on the coating surface.

In the process of friction, two main stages can be distinguished. In the initial period of cooperation, the friction coefficient increased rapidly and, after forming a contact, stabilized in the range of 0.13–0.30. In Figure 9, an example of graphs registered during the friction test is presented. The measured friction coefficient values correspond well with the coating microstructure presented in Figure 3. It is evident that the applied method not only makes evaluation of the friction coefficient possible but also allows the determination of its variability at the cross section of the coating. Thus, in the case of the reference sample (Figure 9a), the transition from the η phase, which was partially chipped during the formation of the disc–pin contact, to the ζ and δ phases resulted in a reduction in the coefficient of friction from the approximated value of 0.30 (η) to approximately 0.23 (ζ + δ). After HT (430 °C), the subsurface coating layer was composed of a mixture of η and ζ phases in different proportions. Therefore, the value of the friction coefficient was much lower—0.14—and decreased slightly to 0.13 towards the steel surface.

#### 3.2.2. The Schatz Analyse Method

A typical graph registered during friction coefficient measurements using the Schatz Analyse M12 testing machine system, type 5413-2777-03 C, is presented in Figure 10, while the achieved results are listed in Table 2. During the test, the following parameters were used: abutting diameter, 15.55 mm; rate, 10 per minute; sensor T/Ang: T, 200 N m, 1033595; sensor F/T_th_: F, 100 kN–150 N m. The following coefficients were measured: µ_b_, the head friction coefficient; µ_th_, the thread friction coefficient; and µ_tot_, the overall friction coefficient. Considering that the heating rate of the tested bolts was much lower than that of the disc-shaped samples, and the coatings had neither cracks nor delamination, the samples treated at 430 °C (2) were used for friction tests. In order to determine the tendency toward changes in the friction coefficient, the samples without treatment (0) and after processing at temperatures of 300 °C (1) and 530 °C (3) were added. 

The achieved results, shown in Table 2, confirm that heat treatment influenced the friction coefficients of zinc coatings deposited on the bolt head and thread in different ways. The μ values differ markedly because the thicknesses and microstructures of the coatings on the bolt head and thread are different. This is mainly due to the difference in heating rate, which is much higher on the thread; therefore, a more intensive diffusion process is to be expected here. Both the µ_b_ and µ_th_ values were lower after heat treatment at 530 °C and 430 °C than after heat treatment at 300 °C. The corresponding total friction coefficients at these levels were µ_tot_ = 0.30 (300 °C), µ_tot_ = 0.24 (430 °C), and µ_tot_ = 0.14 (530 °C). These changes in tendency can follow from the coating hardness increase, as well as from the changes in surface quality (smoothness, degree of surface development, etc.). Nevertheless, the measured values of the friction coefficient are similar to the results achieved with the use of the traditional pin-on-disk method, where measurements were performed on the disc-shaped samples.

## 4. Conclusions

(1)With the use of controlled heat treatment of hot-dip zinc coatings, it is possible to successfully reduce the abrasive wear of zinc coatings applied to steel elements and to adjust/change the coefficient of friction according to the operating instructions, within the range of 0.30–0.14.(2)Microhardness measurements made on the samples subjected to HT in the tested temperature range showed an approximately 3–5 times increase of the HV in the surface layer (145–285 HV 0.02) in relation to the reference samples (58 HV).(3)After HT (at temperatures below 430 °C), the outside surface of the coating essentially did not change (in either topography or coloring). The groove resulting from friction during the pin-on-disc test was regular. The particles rubbed away from the coating had a "flake" shape and size of 0.1–0.6 mm.(4)Despite the slight differences seen during macroscopic verification of the tested coatings heat-treated in the temperature range 300–430 °C, with increasing heat treatment temperature, the coating showed higher abrasion resistance, which was manifested by a reduction in weight loss measured during the tribological test.(5)Heat treatment of zinc coatings at higher temperature (530 °C) caused, besides changes in the microstructure, significant changes in the color of the coating and an essential increase in weight loss due to friction. Additionally, the sheared zinc coating particle shape changed from “flakes” to “fine powder”.(6)Using the Schatz Analyse system, it was quite easy to evaluate the heat treatment’s influence on the friction coefficient of the zinc coatings in different places on the bolts. Both the µ_b_ and µ_th_ values were lower after the heat treatment at 530 °C than after the heat treatments at 430 and 300 °C. This knowledge of parameter value is especially important during automatic assembly.(7)For the proper evaluation of bolt properties, the application of two tests is very useful—the standard pin-on-disk tribometer enables wear mechanism analysis and determination of friction coefficient variability in the cross section of the coating, and the Schatz Analyse system allows technical description of bolt characteristics under realistic conditions.(8)Considering that the size, geometry, and weight of galvanized elements influence HT’s effects on coatings, the individual selection of treatment conditions (temperature and time) should be preceded by detailed analysis of the heating process.

## Figures and Tables

**Figure 1 materials-14-00660-f001:**
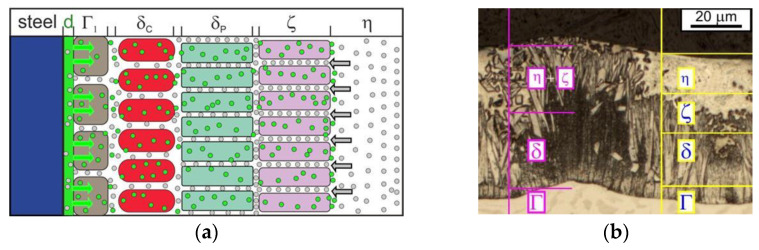
(**a**) The model of ideal zinc layer growth during hot-dip zinc galvanizing [4]; (**b**) the diversity of the microstructure of the zinc coating (cross section) deposited on steel (authors’ own investigation).

**Figure 2 materials-14-00660-f002:**
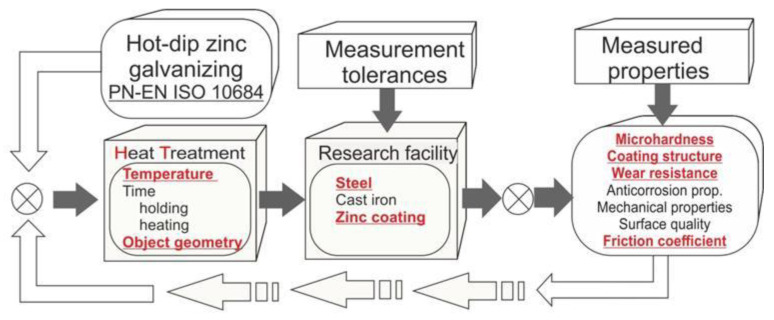
A diagram explaining the conducted experiment.

**Figure 3 materials-14-00660-f003:**
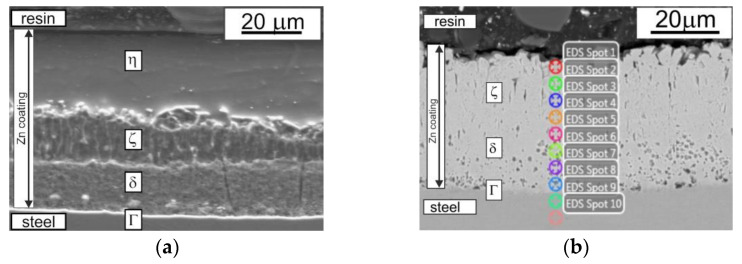
The microstructure observed at the cross section of the zinc coating deposited on disc-shaped samples: (**a**) without HT; (**b**) after HT (460 °C), with marked EDS analysis points.

**Figure 4 materials-14-00660-f004:**
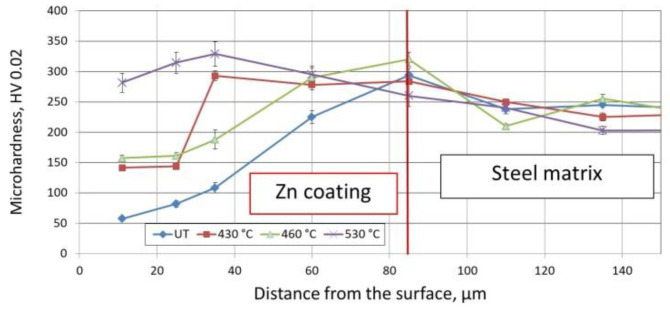
Microhardness measured at the cross section of the heat-treated zinc coating deposited on disc-shaped steel samples (430 °C, 460 °C, 530 °C) in comparison to an untreated (UT) reference sample.

**Figure 5 materials-14-00660-f005:**
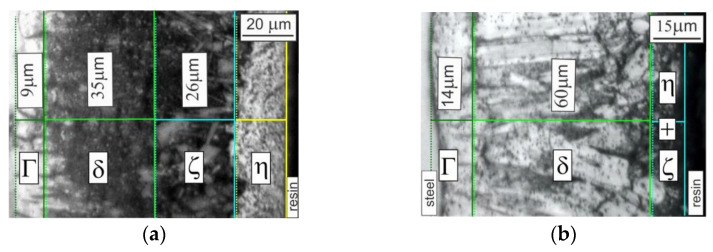
The microstructure observed at the cross section of a zinc coating deposited on a bolt head: (**a**) before HT; (**b**) after HT at 430 °C.

**Figure 6 materials-14-00660-f006:**
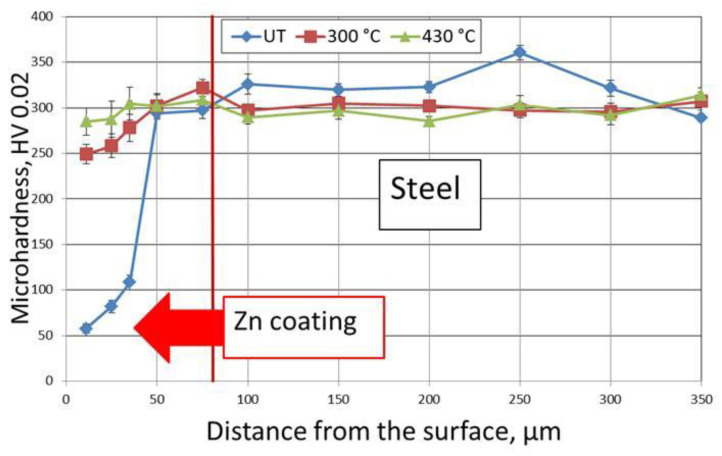
Microhardness measured at the cross section of zinc coatings deposited on M12-60 bolts: untreated and after HT at 300 °C and 430 °C.

**Figure 7 materials-14-00660-f007:**
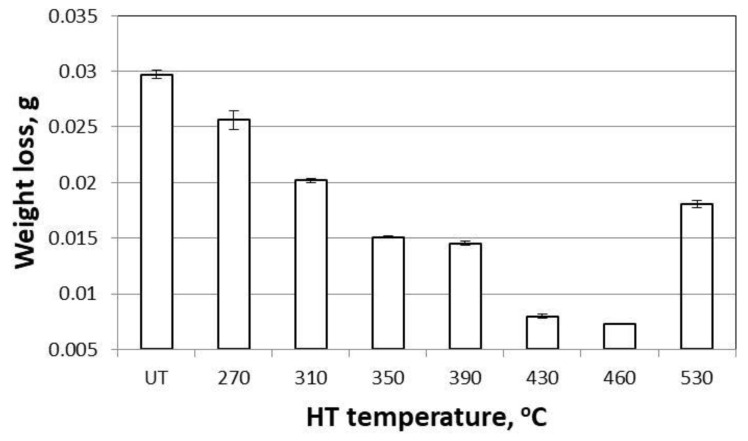
The disc-shaped samples’ weight loss after the pin-on-disc friction test.

**Figure 8 materials-14-00660-f008:**
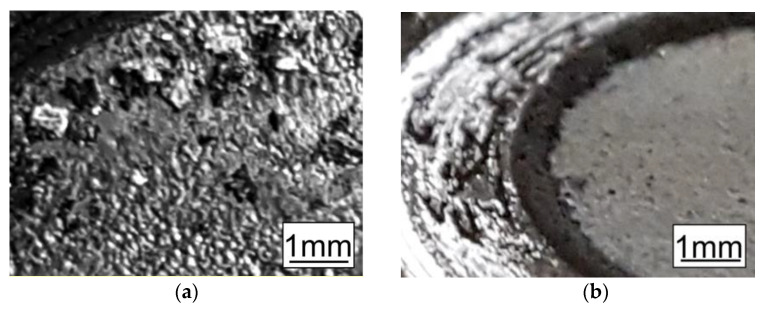
The coatings’ outside surface appearance observed after the pin-on-disc friction test: (**a**) after HT at 270 °C; (**b**) after HT at 530 °C.

**Figure 9 materials-14-00660-f009:**
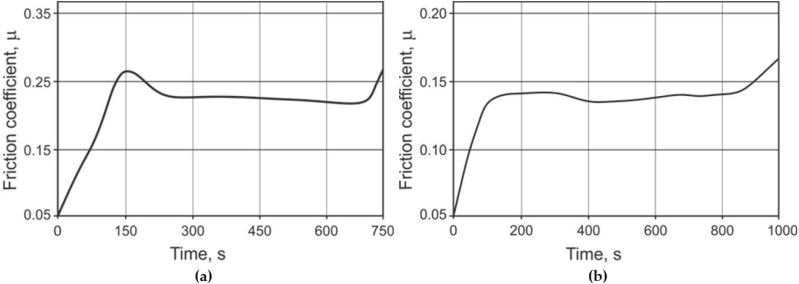
The friction coefficient values registered during pin-on-disc testing of disc-shaped samples: (**a**) without HT; (**b**) after HT at 430 °C.

**Figure 10 materials-14-00660-f010:**
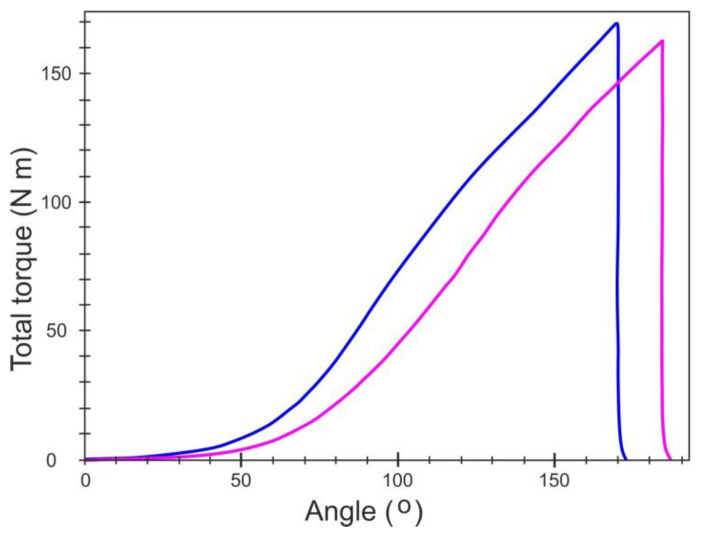
An example graph registered during friction coefficient measurements using the Schatz Analyse M12 testing machine—sample no. 2, heat treated at 300 °C.

**Table 1 materials-14-00660-t001:** The results of EDS analysis of the disc-shaped sample coating heat-treated at 460 °C.

Element	Distance from the Coating Surface, µm							
	5	15	25	30	40	50	60	70
	Content of the Element, wt%							
Fe	6.18	6.47	6.80	7.28	7.95	8.48	12.71	25.56
Zn	93.00	92.90	92.36	91.96	91.31	90.65	86.47	73.39
O	0.82	0.63	0.84	0.76	0.74	0.87	0.82	1.05

**Table 2 materials-14-00660-t002:** The results of friction coefficient measurements.

Sample No.	F (kN)	T (N m)	T_b_ (N m)	T_th_ (N m)	µ_b_	µ_th_	µ_tot_
0.1	36.69	153.63	79.67	73.96	0.28	0.28	0.28
0.2	36.68	158.26	81.29	76.98	0.28	0.29	0.29
0 average	36.69	155.95	80.48	75.47	0.28	0.29	0.29
1.1	36.69	169.54	122.93	46.61	0.43	0.16	0.31
1.2	36.70	161.57	102.12	59.45	0.36	0.21	0.29
1 average	36.69	165.55	112.52	53.03	0.39	0.19	0.30
2.1	36.69	130.87	78.31	52.56	0.27	0.18	0.23
2.2	36.69	139.53	91.78	47.74	0.32	0.16	0.25
2 average	36.69	135.20	85.05	50.15	0.30	0.17	0.24
3.1	36.69	79.54	35.40	44.14	0.12	0.15	0.13
3.2	36.68	84.90	37.20	47.69	0.13	0.16	0.15
3 average	36.68	82.22	36.30	45.92	0.12	0.15	0.14

## Data Availability

Data is contained within the article.

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
