# Peer review of "The Impact of Heat Treatment on the Behavior of a Hot-Dip Zinc Coating Applied to Steel During Dry Friction"

_materials, 2021, doi:10.3390/ma14030660_

Round 1
Reviewer 1 Report
Even though a lot of work has been done to carry out the study, a lot of issues limit the scientific soundness of the paper. In my opinion, the way the data presentation is hard to follow. A lot of details in the description are missing. To consider this work for publishing, the paper should be re-written in a clearer scientific style to fulfill the quality standards of Materials Journal.
Below I gave an example of issues that must be addressed before resubmission, nevertheless the whole manuscript must be corrected accordingly.
Comments to Section :3.1. Metallographic observations and micro-hardness distribution.
- In my view, the authors should present data more clearly and step by step, e.g.; first, discuss the morphology of the coating after only hot-dip galvanization in detail (Figure 3a), and later on move to HT effect discussion.
- SEM should be presented and discussed for all studied temperatures of the HT treatment – it is the key point of this work. It is particularly interesting to see the difference between the samples.
- Line 104: ‘An increase in HT temperature results in an increase in the thickness of the iron-rich layers (G, d)’. – please mark the difference in the pictures. In general, It would make SEM pictures interpretation more clear if the authors mark: coatings layers, the metal substrate and resin/holder (something that does not belong to the sample) on the SEM pictures. The authors also did not describe in the experimental section how samples for cross-section SEM pictures were prepared (samples embedded in resin and polished, or cut somehow?)
- Figure 4.: Celsius degrees are missing in the legend. In line 121 the authors mentioned that “Figure 4 presents the averaged hardness results’’- please add the scale bars. Moreover, in the experiment description the authors did not define to number of repeats.
- Line 107-114: The authors wrote:
‘The outer layer of the zinc coating has a decorative character and improves the susceptibility of covered elements to the plastic processing. On the other hand the outer layer consisting of the phase η corrodes more intensively than intermetallic phases of transitional layer (after wearing of the outside coating layer, the rate of the corrosion process reduces and stabilizes as the layers of intermetallic phases Fe-Zn are dissolve with a lower intensity), so there is a possibility that such a change of the structure will not have a significant influence on the corrosion resistance of the coating.’
In my view in this section, the authors wrote the discussion based on presumptions without evidence supporting these statements. Please limit to conclusions that are based on observation taken from presented data in this manuscript, e.g., the corrosion performance was not verified in this work.
Minor points examples:
- I believe replacing the word ‘deposition’ with ‘electrodeposition’ would give more clarity to this work (both in the title and text).
- The flow of the diagram presented in Figure 2 is hard to understand – I do not follow the logic of this diagram. I would suggest removing it completely.
- Line 78: Could the authors provide more details about the composition of ‘Zn bath with Al, Bi, Ni’?
- Line 80: ‘Next samples were subjected to the controlled heat treatment, in temperature range: 200 - 530 oC (in addition bolts were subjected to processing in temperature 300, 430 and 530 oC).’ From this description it is hard to understand the bolted samples treatment procedure.
- Line 101: ‘The thickness of the coating on the bolts head as well as test samples was every time in range of 80 - 100 µm.’ Which coatings? Galvanized or after HT, or all of them? Additionally, the ‘test samples’ are the ‘disc shaped samples’?
- Line 65: The authors introduce the HT abbreviation for the second time. In my opinion, one time in the abstract is enough.
- Line 76: ‘bolts’ should be placed out of the bracket?
- Line 82: ‘for samples’ please change to ‘for disc shaped samples’ – please be more accurate in the description.
- Line 103: The statement: ‘The changes observed in zinc coating morphology occur in accordance with the Fe-Zn equilibrium diagram [3, 4] and result in an increase coating brittleness.’ is not clear. Could the authors clarify the meaning of this sentence?
- The authors mention in line 106, HT at 460C, in Figure 4 are samples “430C and 470C”. Are the given temperature values correct?
Reviewer 2 Report
Paper presents good technical report evaluating the effect of the heat treatment on the behavior of hot dip zinc coating deposited on steel.
However, some improvements are needed before the paper can be recommended for publication.
HT should refer to Heat Treatment (not to adjective heat treated).
Legends should contain complete information about what is presented in the figures.
Table 1 – crude state (0) is not presented (but it is mentioned in the text).
The comparison between pin-on-disc and Schatz analysis is not well explained (and/or well situated in the text). Best HT is at about 430 áµ’C (according to weight loss measured by pin-on-disc) – how can it be evaluated from Schatz analysis? The comparison obviously holds only for coefficient of friction. Coefficient of friction is closely correlated with hardness but not directly with abrasion resistance. Therefore, this should be written in a more comprehensive way, otherwise the meaning of the whole paper is lost.
Final remark is on the effect of HT on corrosion resistance. No attempt was made to characterize this effect, even if it is mentioned in the introduction (as one of the main aims of the paper). This is a serious shortcoming of the paper.
Reviewer 3 Report
In the article, the authors conducted research on a widely used material such as Zn coating on steel. The material is promising due to its low cost, manufacturability of the method, as well as the high mechanical and anti-corrosion properties of the coating. The use of heat treatment is promising for improving the mechanical and adhesion characteristics, but the effect on anti-corrosion protection is ambiguous. The authors conducted a qualitative, but not exhaustive, experiment. In my opinion, additional studies of Young's modulus and crystal structure of Zn coating would be extremely useful. Nevertheless, the overall rating of the article could potentially be high after corrections and additions in accordance with the comments below.
Therefore, my decision is a major revision.
- Figures 5, 8 and 9 are incomplete. The right side is cut off.
- What is the reason for the choice of heat treatment temperatures? Did the authors build on their previous work? In this case, it will be better to provide relevant Ref. Otherwise, the choice of temperature should be justified.
- In the article, much attention is paid to the microhardness of the coating. The authors showed that after galvanizing and heat treatment, the coating hardness increases. This is an undeniably important and interesting result from a technical point of view. In my opinion, the article does not thoroughly analyze the mechanism of hardness change under the influence of temperature. Probably, the reason may lie in a change in the crystal structure, possibly also in surface oxidation and steel atoms diffusion. Perhaps the authors will be interested to get acquainted with the results obtained by the authors of the article [doi:10.3390/nano10061077. It analyzed the effect of HT at close temperatures on the crystalline structure, microstructure, and mechanical properties of iron-containing coatings. In any case, I think that additional studies of at least the crystalline structure will help to conduct a more complete analysis of the nature of the discovered phenomena.
- It is well known that the hardness does not fully characterize the mechanical properties of coatings if Young's modulus is not also analyzed. A lot of the works, including in-depth analysis of mechanical and tribological properties, investigate the relationship of hardness to Young’s modulus [https://doi.org/10.1007/s11665-018-3483-7, https://doi.org/10.1016/j.vacuum .2018.07.017]. Has any research of Young’s modulus been conducted? If not, I think you should rely at least on the results of other studies.
- In the conclusions, the authors write about a change in the color of the coating after heat treatment at 530 C. Is it due to surface oxidation, diffusion of steel atoms, or is there another reason?
Round 2
Reviewer 1 Report
Although I see a significant improvement in the work quality, still it is not acceptable for publishing in the current form, and the next round of corrections must be done carefully. The writing style is not careful and the authors again skip some important detail in the description. Most importantly, it is not well convincing why the authors selected these samples for testing, and why different tests present results for different samples (different treatment temperatures).
Major issues:
1) Why the treatment temperatures considered in the different tests are varied? For example, in the hardness test (Fig 4) and friction coefficient (Fig. 7) different temperatures are present (first 300C, second 310 C) and different ranges tested. Moreover, why in such a case, the authors did not test 270C sample for hardness? Especially that the authors claim that the tested coating in the temperature range of 300-530 C show too high hardness. The same problem is for the bolts: microhardness tested at 0, 300, 430C while Schatz test is done for 300, 430, and 530C. This lack of consistency unable to make conclusions from the work.
2) Authors: “Taking into consideration the fact that the target is the hardness increase to the level 100-150 HV 0.02, the applied parameters, both temperature and time appear to be too high. The achieved relatively high hardness level can result in too high brittleness of zinc coating.”
- In such a case, could the authors clarify why they continued investigations with these samples, not try to change treatment conditions?
Minor:
3) Figure 1. Scale-bar missing
4) Line 110: Thank you for adding the description of the samples preparation, but please clarify whether the samples were cut and embedded in resin for all tests or only for SEM?
5) Figure 4, Figure 6, Line195: Ascribing untreated sample as 0 C makes some confusion. I suggest to use instead “untreated”, “as-coated” or similar, because I believe the samples were not cooled down up to 0C, but kept at RT.
6) Lines 121-125: Authors; “Taking into account the fact that the thickness of the tested coating (80-100 μm) is much greater than in the compared study (3μm), limitation to micro-hardness measurement at the present stage of the experiment seems to be justified.”
- Can the authors give a reference to the “compared study”.
- I not very clear to me why based on that statement the “limitation to micro-hardness […] seems to be justified”. Please rewrite in a more straightforward way.
7) Line 151: Please rewrite: “460C, reheat after cooling to 500C”. It sounds like cooling was from 460C to 500C (not possible), and reheat was later on.
8) Line 197: Test samples = disc??
9) The section from line 197 is very not clear in my view:
Authors: ”Unfortunately hardness of subsurface steel layer slightly decreases from 320 to 300 and 290 HV 0.02 as an effect of tempering.”
- Please describe more clearly which sample and data point are discussed here. It is hard to understand.
Authors: ”The measurement of HBW hardness in the depth of 2 mm confirmed the hardness decreasing correspondingly from 271 to 255 and 245 HBW.”
- Please rewrite in a more clear way. If you write “from 271 to 255 and 245”, it looks like some values range for one sample.
Authors: ”No delamination, cracks, and surface degradation visible as the result of conducted heat treatment of zinc coating were observed.”
- Is it observation by eye? This statement is not strongly proved without microscopy analysis. Please try to rewrite, at least to pronounce that it is visual observation.
10) Microhardness (line 174) and Pin on the disc method (218)- please add to the text information about sample type-tested (not only in the Figure caption).
11) Line 236: What the authors mean by “outside” surface? It was exposed to the friction test or not exposed? Please add.
12) Line 242. Is it better to obtain “flake” or “fine powder”? And why the authors show the 430C sample in Figure 9. Is it the best one in the authors' view? Please briefly explain in the text.
13) Line 245: Can the authors provide a table with obtained values of EDS results? It is not enough to mention one value (6%) for one sample. How much was before HT? How other elements changes? The authors took the conclusion that it is a reason for “color change” and “the presence of ζ phase”, which is not enough supported by evidence.
14) Line 255, 258: “Eta” ->symbol
15) 287: “pin on disk - where 286 measurements were performed on the head of zinc galvanized bolts.”
- Based on the pin on disk experiment description I understood that the test was done on disc-shaped samples, but from this sentence, it seems to be done on bolts samples. Please clarify.
Reviewer 2 Report
The paper improved and can be recommended for publication in the present state.
The authors states in the response that reviewers' comments are contradictory (underlined!), which, however, means that they didn't get the point of the comments. To remove (or better to say not to amend) the section on the corrosion resistance is an easy way, unfortunately reducing the scientific quality of the paper…
Author Response
Dear Reviewer No. 2.
Thanks a lot for all your comments and final positive opinion. I am attaching the revised text of the article again with changes according the additional requirements of the Reviewer No.1. The language correctness of the new text has been verified by the English teacher.
Thanks again for your understanding for the removal of the anticorrosion paragraph.
Reviewer 3 Report
I'm completely satisfied. I feel that revised version can be accepted.
Author Response
Dear Reviewer No. 3.
Thanks a lot for all your comments and final positive opinion. I am attaching the revised text of the article again with changes according the additional requirements of the Reviewer No.1. The language correctness of the new text has been verified by the English teacher.
Round 3
Reviewer 1 Report
I am satisfied with the correction done on the manuscript and I can recommend it for publishing in the current form.